# Cost-Effectiveness of Therapeutic Drug Monitoring of Anti-TNF Therapy in Inflammatory Bowel Disease: A Systematic Review

**DOI:** 10.3390/pharmaceutics14051009

**Published:** 2022-05-07

**Authors:** Silvia Marquez-Megias, Ricardo Nalda-Molina, Javier Sanz-Valero, Patricio Más-Serrano, Marcos Diaz-Gonzalez, Maria Remedios Candela-Boix, Amelia Ramon-Lopez

**Affiliations:** 1School of Pharmacy, Miguel Hernández University, 03550 San Juan de Alicante, Spain; silvia.marquez@goumh.umh.es (S.M.-M.); mas_pat@gva.es (P.M.-S.); aramon@umh.es (A.R.-L.); 2Alicante Institute for Health and Biomedical Research (ISABIAL-FISABIO Foundation), 03010 Alicante, Spain; diaz_marcosgon@gva.es; 3Department of Public Health and History of Science, School of Medicine, Miguel Hernandez University, 03550 Alicante, Spain; fj.sanz@isciii.es; 4Carlos III Health Institute, National School of Occupational Medicine, 28029 Madrid, Spain; 5Clinical Pharmacokinetics Unit, Pharmacy Department, Alicante University General Hospital, 03010 Alicante, Spain; 6Virgen de la Salud General Hospital of Elda, 03600 Elda, Spain; candela_marboi@gva.es

**Keywords:** inflammatory bowel diseases, drug monitoring, pharmacokinetics, tumor necrosis factor inhibitors, adalimumab, infliximab, cost–benefit analysis, cost-effectiveness

## Abstract

Infliximab and adalimumab are monoclonal antibodies against tumor necrosis factor (anti-TNF) used to manage inflammatory bowel disease (IBD). Therapeutic Drug Monitoring (TDM) has been proven to prevent immunogenicity, to achieve better long-term clinical results and to save costs in IBD treatment. The aim of this study was to conduct a systematic review on cost-effectiveness analyses of studies that apply TDM of anti-TNF in IBD and to provide a critical analysis of the best scientific knowledge available in the literature. The quality of the included studies was assessed using Consolidated Health Economic Evaluation Reporting Standards (CHEERS). Cost-effectiveness of the TDM strategies was presented as total costs, cost savings, quality-adjusted life-years (QALY) and incremental cost-effectiveness ratio (ICER). Thirteen studies that examined the health economics of TDM of anti-TNF in IBD from 2013 to 2021 were included. Eight of them (61.5%) achieved a score between 17 and 23 on the CHEERS checklist. The comparison between the TDM strategy and an empirical strategy was cost saving. The ICER between reactive TDM and an empirical strategy was dominated (favorable) by reactive TDM, whereas the ICER value for proactive TDM compared to an empirical strategy ranged from EUR 56,845 to 3,901,554. This systematic review demonstrated that a TDM strategy is cost-effective or cost-saving in IBD.

## 1. Introduction

Crohn’s disease (CD) and ulcerative colitis (UC) are autoimmune inflammatory bowel diseases (IBD) characterized by the chronic inflammation of the intestinal tract [1,2]. In 2017, there were 6.8 million cases of IBD globally [3,4,5]. According to several European studies, the risk of colorectal cancer is two times higher in UC patients than in the general population, whereas patients with CD have a higher risk of extraintestinal cancers and increased mortality compared to the general population [6,7,8,9,10].

Infliximab and adalimumab are monoclonal antibodies against tumor necrosis factor (anti-TNF), which are increasingly used to treat patients with moderate-to-severe IBD older than 6 years who had an inadequate response with corticosteroids or immunomodulators [11,12]. However, nearly 40% of IBD patients experience a loss of response (LOR) to anti-TNF treatment every year, requiring either dose intensification or switching to another drug [13].

Numerous studies reveal that higher serum drug concentrations are associated with better therapeutic outcomes, including mucosal healing [14,15,16,17,18]. Related to this, some authors have shown that patients can experience LOR to anti-TNF due to developing antibodies against anti-TNF (AAA) after long periods of subtherapeutic drug levels [19,20,21,22]. In this line, Therapeutic Drug Monitoring (TDM) of anti-TNF has been proven to prevent immunogenicity and to achieve better long-term outcomes in terms of IBD-related surgery or hospitalization [23,24,25]. Since the mean direct cost per patient-year is around EUR 3500 for CD patients and EUR 2000 for UC patients including diagnostic procedures, hospitalizations and biological treatment in Europe [26], TDM could be a tool of special relevance to optimize the treatment and save costs.

In the literature, other systematic reviews confirmed that TDM of anti-TNF is cost-effective in rheumatoid arthritis [27,28]. Recently, another systematic review of the TDM of immunomodulators and anti-TNF therapy in IBD proved to be cost-effective or cost-saving compared with an empirical strategy without TDM. However, the main limitation of this review is the inclusion of only model-based analyses with simulations of patients [29].

The aims of this systematic review are to evaluate studies concerning the cost-effectiveness analysis of TDM of anti-TNF in IBD and to provide a critical analysis of the best scientific knowledge available on the use of TDM.

## 2. Materials and Methods

### 2.1. Design

This systematic review consisted of a cross-sectional descriptive study and a critical analysis of studies found in the literature. The structure of this review followed the Preferred Reporting Items for Systematic Reviews and Meta-Analyses (PRISMA) guidelines [30] (Appendix A), and the methodological framework proposed for scoping studies [31,32].

### 2.2. Source of Data Collection

The data were obtained from direct consultation and access, via the Internet, to the following bibliographic databases in the field of health sciences: MEDLINE (via PubMed), Embase, Cochrane Library, PsycINFO, Scopus, Web of Science, and Latin American & Caribbean Health Sciences Literature (LILACS) and Medicina en Español (MEDES). The published articles were analyzed and retrieved from the indicated bibliographic databases.

### 2.3. Information Processing

Search terms were selected using the Thesaurus of Health Sciences Descriptors (DeCS) developed by the Latin American and Caribbean Center on Health Sciences Information (BIREME) and equivalent terms established by the US National Library of Medicine, Medical Subject Headings (MeSH). The MeSH descriptors “Inflammatory Bowel Diseases”, “Tumor Necrosis Factor Inhibitors”, “Infliximab”, “Adalimumab”, “Cost-Benefit Analysis”, “Cost Savings” and “Drug Monitoring” were considered suitable. Likewise, these terms were used to query the database using the title and abstract field (Title/Abstract). It was not necessary to use filters (limits). The search equations are available in Appendix A. This review was not registered; although, the protocol was developed before the research began.

This strategy was subsequently adapted to the characteristics of each database consulted, from the first available date in each of the selected databases until December 2021. Additionally, a complimentary search strategy was performed to reduce the possibility of publication bias by manually searching the reference lists of the articles that were selected for the review. Likewise, experts in the subject under study were contacted to determine the possible existence of gray literature (materials and research produced by organizations outside traditional commercial or academic publications that are disseminated through other distribution channels).

### 2.4. Final Selection of Articles

For the review and critical analysis, articles that met the following criteria were chosen:

Inclusion: original articles published in peer-reviewed journals and articles that met the objectives of the search.

Exclusion: full text could not be found; no relationship between the intervention and the outcome under study (causality criterion); articles that included any diseases different to IBD such as rheumatoid arthritis, psoriasis or ankylosing spondylitis; studies regarding different drugs to anti-TNF antagonists such as Vedolizumab or Ustekinumab and studies developed in animals.

There was no language, publication date or publication status restriction. The selection of relevant articles was performed by two authors of the present review (S.M.-M. and A.R.-L.). To validate the inclusion of the articles, the assessment of the agreement between the authors using the kappa index, had to be greater than 0.60 [33]. In case of discrepancies, a third reviewer (R.N.-M.) was responsible for reaching a resolution and subsequent consensus amongst all the authors.

### 2.5. Quality Assessment, Level of Evidence and Grade of Recommendation

The quality of all identified studies was evaluated by the Consolidated Health Economic Evaluation Reporting Standards (CHEERS) checklist [34], which consists of 24 items to appraise the quality of the included studies. For each item, it was assigned one point for each item present (if not applicable, it was not scored). The percentage of total scores assigned to each study was used to evaluate the study quality. All studies were classified into four categories: “excellent” (≥85%), “very good” (70–84%), “good” (55–69%), and “insufficient” (<55%).

The quality assessment was conducted by two reviewers (S.M.-M. and A.R.-L.) independently. It was established that the inter-rater agreement for the authors (using the kappa index) should be higher than 60%. In case of discrepancies, a third reviewer (R.N.-M.) was responsible for reaching a resolution and subsequent consensus amongst all the authors.

To assess the risk of bias due to missing results, the methods and the results sections of the selected articles were compared. To determine the level of evidence and its degree of recommendation, the recommendations of the Scottish Intercollegiate Guidelines Network Grading Review Group (SIGN) [35] were used.

### 2.6. Data Extraction

Data from eligible articles were collected to systematize and facilitate the interpretation of the results. Data were extracted by one reviewer and verified by a second reviewer. The following items were extracted: general information of study (first author, country of the study and year of publication); study design (population, intervention, type of TDM approach used, time horizon and methods of measuring outcomes); and results (primary outcomes). The primary outcomes collected were the cost-effectiveness of the TDM strategies, presented as the total costs, cost savings, quality-adjusted life-years (QALY) or incremental cost-effectiveness ratio (ICER). Costs were converted to the same currency (euros) in the year of publication of each study to allow the comparison between the different studies. Moreover, all the costs were normalized to a one-year period to homogenize the results. If the information was available, average costs per patient per year were calculated according to the total costs and the number of patients in each group of treatment in each study.

## 3. Results

This systematic review identified a total of 102 publications: 33 were found in Medline (via PubMed), 16 in Embase, 1 in Cochrane Library, 18 in Scopus and 34 in Web of Science. No article was retrieved from PsycINFO, LILACS and MEDES. A total of 13 original articles were included in this review [36,37,38,39,40,41,42,43,44,45,46,47,48] after removing duplicates, applying the inclusion and exclusion criteria and consulting the bibliographic lists of the selected articles from the search strategy. The list of excluded studies and the reasons for exclusion is available in Appendix A. The inter-rater agreement for the selected studies was 0.815 (*p* < 0.001) according to the kappa index.

The process of study selection is presented in a flowchart in Figure 1.

All included studies examined the health economics of TDM in IBD patients treated with anti-TNF from 2013 to 2021. Moreover, 12 of them evaluated the TDM of infliximab and only one (ST1) evaluated the use of adalimumab. The summary of the characteristics of each study is listed in Table 1.

### 3.1. Study Design

Regarding the study design, three were randomized controlled clinical trials (ST11–13), one was a non-randomized clinical trial (ST10), two were prospective observational studies (ST7, 9) and one was a retrospective observational study (ST8).

Alternatively, six studies considered simulated patients using a modeling approach, based on a Markov model (ST1, 4, 6), a stochastic simulation model (ST2) and a discrete event model (ST3, 5).

### 3.2. Population

Twelve studies were carried out on adult patients (ST3, 10, 4–6, 11–13) while only one studied the pediatric population (ST1). Eight studies considered only CD patients (ST1–4, 6, 11–13), whereas five included IBD patients (CD and UC) (ST7–11). Eight studies included patients only in a maintenance phase of treatment (ST3, 4, 6, 7, 9, 10, 12, 13), and in the others, patients were in both induction and maintenance stages.

### 3.3. Interventions

The included studies show three types of intervention: proactive TDM, reactive TDM and an empirical strategy. Proactive TDM can be defined as the measurement of concentrations and AAA levels in all patients, at specific time points, to optimize drug dosage and to achieve a threshold drug concentration that can improve response rates and prevent secondary LOR and the development of AAA [49]. Reactive TDM is the measurement of drug concentration and antibody levels only when patients had experienced primary or secondary LOR to a biological treatment to inform about reasons for the lack of response and to facilitate the next therapeutic decisions such as increasing drug, adding immunomodulators or switching to another drug either in or out of class [49]. Another approach is an empirical strategy that bases its dosage changes on clinical symptoms. Four compared a proactive TDM versus a reactive TDM (ST1, 7, 8, 10), three compared a proactive TDM versus an empirical strategy (ST3, 4, 11), five compared a reactive TDM versus an empirical strategy (ST5, 6, 9, 11–13) and one study included the three strategies as part of its intervention (ST2).

### 3.4. TDM

All studies, either proactive or reactive TDM, applied an algorithm to decide the next decision in the treatment based on drug concentrations. Three studies developed and used their own algorithm (ST2, 3, 12) while the rest adapted or used others found in the literature. Five of the algorithms included an optimal drug concentration interval (ST1, 4, 7, 10, 11), five considered a threshold (ST3, 5, 9, 12, 13), whereas three did not mention the interval or threshold used to change the dosage or to switch drug (ST2, 6, 8). Seven algorithms took into account the use of immunomodulators (ST1, 2, 4, 5, 9, 12, 13); although, four of them did not include their cost in the analysis (ST5, 9, 12, 13).

Six studies defined the clinical or biochemical criteria to establish response (ST6, 9–13) while four described the criteria to determine LOR (ST1, 9, 12, 13). Even though they were distinct from each other, all of them included the Crohn’s Disease Activity Index to determine either response or LOR. In relation to LOR, four out of five studies that included induction patients considered the primary LOR and the discontinuation of the drug due to adverse events (ST1, 2, 5, 11). Regarding secondary LOR, three studies (ST1, 5, 6) included the cost of the treatment with another drug after switching from the main drug, whereas one study (ST8) did not include this cost in the final result.

The analytical assay to measure AAA and trough concentrations differed in each study: five (ST3–5, 7, 9) used LISA TRACKER duo (Theradiag, Marne la Vallée, France); one (ST8) IDK monitor ELISA kit; one (ST10) homemade ELISA (Sanquin, Amsterdam, The Netherlands); two (ST12, 13) radioimmunoassay (Biomonitor A/S, Copenhagen, Denmark); and the rest did not specify it. In addition to this, three studies only measured AAA if trough concentrations were below an established threshold (ST7, 10, 11), one study only measured trough concentrations in absence of AAA (ST6), whereas the others tested the levels of both AAA and trough concentrations simultaneously at every measurement, which considerably affected the cost.

### 3.5. Costs

The costs analyzed varied from each study. The outcomes were measured as costs in nine studies (ST1–4, 6, 9, 11–13), as cost savings in all included studies, as QALY and as ICER in five studies (ST1, 2, 4, 6, 11). Eight studies (ST1–4, 6, 9, 12, 13) included an extensive detailed amount of costs such as clinic visits, hospitalization, surgery and diagnosed tests, among others. The cost of the anti-TNF treatment was evaluated in all of the included studies; although, in some studies patients were treated with biosimilars such as Inflectra (ST8), CT-P13 (ST3, 9) and another study (ST7) did not specify it. Attar et al. (ST3) defined two scenarios to calculate the cost of the treatment with all patients treated with the originator (Remicade) and with the biosimilar CT-P13, whereas Guidi et al. (ST9) added a third scenario to calculate half patients treated with each one. The summary of the costs of each study is listed in Table 2.

Total costs ranged from EUR 14,927 to 186,635,650 per year for a proactive TDM strategy; from EUR 14,263 to 3,230,810 per year for a reactive TDM strategy; and from EUR 14,268 to 201,879,000 per year for an empirical strategy. In this line, cost savings of proactive TDM compared to reactive TDM ranged from EUR 558 to 196,394 per year; cost savings of proactive TDM compared to an empirical strategy ranged from EUR 1391 to 15,243,350 per year; and cost savings of reactive TDM compared to an empirical strategy ranged from EUR 5.39 to 26,260,059 per year (Table 2).

On the other hand, QALY ranged from 0.63 to 0.82 for a proactive TDM strategy; from 0.73 to 0.80 for a reactive TDM strategy; and from 0.65 to 0.84 for an empirical strategy. The ICER between reactive TDM and an empirical strategy was dominated (favorable) by reactive TDM in two studies (ST2, 6). When it comes to ICER between proactive TDM and reactive TDM, the proactive TDM dominated in one study (ST1) while in another study (ST2) its value was EUR 131,858 (below the cost-effective thresholds in the United States). The ICER between proactive TDM compared to an empirical strategy ranged from EUR 56,845 to 3,901,554.

### 3.6. Quality Assessment

The total score of the CHEERS checklist of each study is included in Table 1 and more details are available in Appendix A. The scores ranged from 4 to 23. The number of studies categorized as “excellent”, very good” and “insufficient” was 2 (15.4%), 6 (46.2%) and 5 (38.4%), respectively.

The inter-rater agreement for the determination of each score was 0.806 (*p* < 0.001) according to the kappa index. No risk of bias was observed in the published papers.

Furthermore, following the recommendations of the Scottish Intercollegiate Guidelines Network Grading Review Group (SIGN) [35], the level of evidence was 2++ (high-quality case-control or cohort studies with a very low risk of confounding, bias or chance and a high probability that the relationship is causal) and its degree of recommendation was B (studies rated 2++ directly applicable to the target population).

## 4. Discussion

The objective of this systematic review was to identify and synthesize the scientific evidence published around the cost-effectiveness analyses of the use of TDM of anti-TNF in IBD. This review includes both model-based and trial-based studies. With the object of minimizing publication bias, the database searches were exhaustive, with neither language nor date restrictions. Moreover, the PRISMA guideline was followed to minimize bias and the quality of identified studies was evaluated with the CHEERS checklist. The results of the CHEERS checklist show that eight of thirteen (61.5%) included studies achieved very good to excellent rankings.

After performing an exhaustive search in numerous databases, thirteen studies were found in the literature. Six of them (ST1–6) used a modeling approach based on the calculation of probabilities of having a flare or being included in a TDM strategy, obtained from the literature. A modeling approach allows for the evaluation of large cohorts of patients, which would be very difficult to acquire in real life. However, the main weaknesses of these studies are the simplification of the events related to disease progression, the reliability of the external clinical results used for the modeling and the difficulties in predicting or reflecting a clinical setting.

With regard to the drug studied, 12 studies were focused on infliximab. Therefore, there is a lack of evidence on the cost-effectiveness of using TDM of adalimumab since only one study (ST1) found in the search evaluated the costs with a modeling approach. Due to the limitations of the modeling approach, further studies, either clinical trials or observational studies, are needed to provide a wider outlook.

Five studies (ST6, 9, 10, 12, 13) reported data with a follow-up lower than or equal to 1 year but, as IBD is a chronic disease, a higher follow-up is required to understand the long-term impact on the costs of a TDM strategy [44,50].

Regarding the population, there is only one published study (ST1) based on a pediatric population. Moreover, this study approximated the utility values of health states from studies on adult patients. For that reason, future cost–benefit analyses in pediatric populations are needed to confirm the results of this study.

None of the selected studies recruited UC patients separately from CD and, consequently, the cost-effectiveness evaluation of TDM in UC is lacking. In fact, there may be differences in the response to infliximab between CD and UC patients since infliximab could be more immunogenic and reach lower trough concentrations in UC patients than in CD patients, affecting considerably the cost-effectiveness of TDM of anti-TNF in this group of patients [43].

It has been shown that AAA are clinically relevant for disease progression or applying TDM [19,20,21,22]; however, one study (ST3) did not include the presence of AAA to infliximab because of the complexity generated in its model. Moreover, AAA to infliximab frequently appear in the first year [39]; although, most studies included patients just in a maintenance phase of treatment. Therefore, TDM including the induction of the treatment could lead to more benefits and be more cost-effective since it would prevent flares and decrease hospitalizations and surgery rates [45,51,52,53].

Concerning the intervention, a proactive TDM strategy was included as one of the interventions in eight studies (ST1–4, 7, 8, 10, 11). However, some of them (ST3, 7, 10) applied this concept differently and interpreted it as a strategy for patients that had disease remission and only measured drug trough concentrations and/or AAA levels once. Other authors (ST1, 2, 4, 8, 11) repeatedly measured these levels to avoid LOR and to restrain the disease and its clinical symptoms. In fact, an adequate sample schedule has not been established so far. Consequently, every one of these studies differed in that schedule and used either 8 weeks (ST1), 6 months (ST2) or 1 year (ST4). Therefore, the comparison of the results is difficult due to the lack of generalizability and the fact that a high frequency of TDM could increase the total cost of a proactive TDM strategy as was observed by Yao et al. [36].

The progression of IBD is difficult to predict using a standardized algorithm because different and random events may occur along a patient’s disease course and differ from one patient to another. Conversely, each study applied a different algorithm to achieve a dosage optimization with variations in the interval or threshold used. In this context, patients can be switched to different groups of treatment when applying different algorithms and this could considerably limit the generalizability and bias the overall costs. Moreover, there is no homogeneity in the decisions taken by every algorithm since five studies (ST2, 6, 9, 12, 13) did not consider supratherapeutic drug concentration and, therefore, increasing the dosage was the only possible decision in their algorithm. On the contrary, other studies (ST3, 7) took into account only high drug trough concentrations and, therefore, increasing the dosage was not an option. So far, the exposure target for anti-TNF is highly dependent on the therapeutic objective (clinical, endoscopic, biochemical or histologic remission) and whether patients are diagnosed with CD or UC [17]. Based on the currently available evidence, an interval of 6–10 mg/L of infliximab trough concentrations is recommended to achieve clinical response [25,54,55]. Regarding adalimumab, a target of 8–12 mg/L adalimumab trough concentrations is required to achieve mucosal healing in 80–90% of IBD patients [25,54,55]. In fact, a recent study showed that patients with adalimumab trough concentrations lower than 8.3 mg/L had more risk to develop AAA and to experience LOR by week 12 [56]. The generalization of the clinical use of these intervals would allow the therapeutic decisions and to compare costs to be standardized.

Related to the dosages, 12 studies (ST1, 3–13) administered doses of 5 mg/kg and 10 mg/kg. Nevertheless, Negoescu et al. (ST2) included a medium dose of 7.5 mg/kg of infliximab as an option, and observed that smaller dose increases would decrease the overall cost of the drug and still achieve therapeutic trough concentrations [39].

Recently, adding immunomodulators to the anti-TNF therapy has shown clinical relevance in the decrease or disappearance of AAA [39,57]. However, not all studies included this option in their algorithm (ST3, 6–8, 10, 11) and some of those that included the algorithm did not consider its costs (ST5, 9, 12, 13), which could considerably affect the cost-effectiveness analysis.

The analytical assay varied from each study and, consequently, the limit to consider undetectable AAA and trough concentrations and its cost differed largely from each other. The selected intervals or thresholds are directly dependent on assay type and whether or not AAA are measured. However, none of the assays can be classified as a gold standard. Moreover, four studies (ST1, 2, 6, 11) did not mention which analytical assay was applied, which is essential due to its direct impact on the cost-effectiveness of any TDM strategy.

The definitions of clinical response and LOR are essential to classify patients into different groups and to decide their future treatment. However, seven studies (ST1–5, 7, 8) did not define the criteria to determine a clinical response to the drug and nine studies (ST2–8, 10, 11) did not define the criteria to establish LOR. In those that included a definition for either clinical response and LOR, there are significant variations between them across the studies that have large implications for the generalizability of study outcomes. Moreover, some studies (ST1, 2, 4, 6) carried out procedures such as endoscopies or colonoscopies that significantly increased the overall costs.

Regarding the economic outcomes, it is difficult to compare the results given the different number of patients, the number of drug or AAA samples per patient and the different items included in the calculation of the final costs of each study. The overall costs are closely related to the drug cost, but it is not homogeneous and varied between countries and health systems. Furthermore, all selected studies only included direct medical costs of health care such as drug cost, drug or AAA testing, surgery and hospitalization. As none of them considered the indirect costs associated with flaring (e.g., time missed from work) and the likelihood of experiencing an adverse event, the economic implications of TDM might be underestimated. Moreover, biosimilars would enhance a more cost-effective strategy due to their lower price [58,59]; although, all the selected studies concluded that the TDM strategy is cost-effective compared to an empirical strategy.

All included studies used TDM to optimize the treatment of anti-TNF in IBD patients through an algorithm. According to Papamichael et al., TDM was defined as the evaluation of drug concentration and anti-drug antibodies [60]. However, Model-Informed Precision Dosing (MIPD) is a more precise alternative, based on the use of population pharmacokinetics (PopPK) models and a prospective Bayesian approach to increase the homogeneity in the drug exposure in patients and, therefore, to improve outcomes of treatments [61]. Some authors have carried out MIPD of adalimumab and infliximab in IBD patients, applying PopPk models found in the literature [62,63]. However, further investigations in this line are required to estimate its cost and compare it with the ones obtained from a TDM strategy.

One limitation of this systematic review could be the high rate of non-relevant articles (108) in relation to the final selection made (13). Scopus and Web of Science databases initially retrieved many works that were finally irrelevant, which could be due to the lack of indexing (the search was performed in text format querying the title, abstract and keywords). Despite performing a comprehensive search, it cannot be ruled out that some studies were not identified by the bibliographic databases searched or the manual search. Another limitation of the present review relies on the fact that a meta-analysis was not performed owing to clinical and methodological heterogeneity such as different study designs, significant differences in the algorithms applied and follow-up periods, variability in definitions of clinical criteria and different TDM approaches applied.

## 5. Conclusions

In conclusion, this systematic review identified 13 health economic studies, eight of which were very good to excellent quality work per the CHEERS checklist.

The comparison between the TDM strategy and an empirical strategy was cost saving. The ICER between reactive TDM and an empirical strategy was dominated (favorable) by reactive TDM, whereas the ICER value for proactive TDM compared to an empirical strategy ranged from EUR 56,845 to 3,901,554. This systematic review demonstrated that TDM of anti-TNF drugs is a cost-effective or cost-saving tool in the management of IBD.

## Figures and Tables

**Figure 1 pharmaceutics-14-01009-f001:**
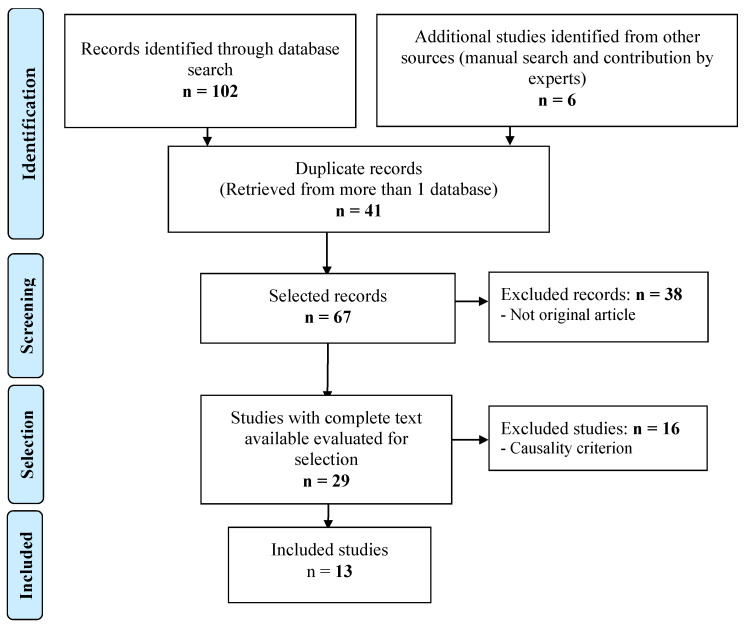
Selection procedure of the studies.

**Table 1 pharmaceutics-14-01009-t001:** Summary of the characteristics of the included studies.

Study	Authors, Year	Country	Study Design	Drug Studied	Study Population (*n*, m/f, Age)	Study Duration	Measure of Outcomes	Intervention	CHEERS (*n*, %)
ST1	Yao et al., 2021 [36]	US	Modeling approach	Adalimumab	20,000 simulated CD pediatric biologic-naïve patients	3 years and 4 weeks	Costs Cost savings QALY ICER	Proactive TDM (*n* = 10,000) vs. Reactive TDM (*n* = 10,000)	21 (87.5%) Excellent
ST2	Negoescu et al., 2020 [37]	US	Modeling approach	Infliximab	100,000 CD simulated patients	5 years	Costs Cost savings QALY ICER	Proactive TDM (*n* = NA) vs. Reactive TDM (*n* = NA) vs. Empirical strategy (*n* = NA)	19 (79.2%) Very good
ST3	Attar et al., 2019 [38]	France	Modeling approach	Infliximab	40,000 CD simulated adult patients	2 years	Costs Cost savings	Proactive TDM (*n* = 20,000) vs. Empirical strategy (*n* = 20,000)	7 (29.2%) Insufficient
ST4	Freeman et al., 2016 [39]	UK	Modeling approach	Infliximab	Simulations of CD patients in maintenance treatment of infliximab	10 years	Costs Cost savings QALY ICER	Proactive TDM (*n* = NA) vs. Empirical strategy (*n* = NA)	23 (95.8%) Excellent
ST5	Roblin et al., 2015 [40]	France	Modeling approach	Infliximab	10,000 Simulations of CD patients with LOR to infliximab	1, 3 and 5 years	Cost savings	Reactive TDM (*n* = NA) vs. Empirical strategy (*n* = NA)	10 (41.7%) Insufficient
ST6	Velayos et al., 2013 [41]	US	Modeling approach	Infliximab	10,000 simulations of CD patients with LOR to infliximab	1 year	Costs Cost savings QALY ICER	Reactive TDM (*n* = NA) vs. Empirical strategy (*n* = NA)	17 (70.8%) Very good
ST7	Wu et al., 2021 [42]	Australia	Prospective observational study	Infliximab	428 IBD patients (322/296, 36 ± 18.7 yo)	56 weeks	Cost savings	Proactive TDM (*n* = 181) vs. Reactive TDM (*n* = 247)	12 (50.0%) Insufficient
ST8	Ganesnanthan et al., 2020 [43]	UK	Retrospective observational study	Infliximab	85 IBD patients (54/31, 39.13 ± 14.25 yo)	NA	Cost savings	Proactive TDM (*n* = NA) vs. Reactive TDM (*n* = NA) vs. Proactive TDM post-induction (*n* = NA)	7 (29.2%) Insufficient
ST9	Guidi et al., 2018 [44]	Italy	Prospective observational study	Infliximab	148 IBD patients in treatment for at least 4 months with LOR to infliximab (75/73, 40.8 (37.05–42.5) yo)	12 weeks	Costs Cost savings	Reactive TDM (*n* = 96) vs. Empirical strategy (*n* = 52)	17 (70.8%) Very good
ST10	Taks et al., 2017 [45]	The Netherlands	Non-randomized clinical trial	Infliximab	33 IBD adult patients (20/13, 43 (32–59) yo)	1 year	Cost savings	Proactive TDM (*n* = 28) vs. Reactive TDM (*n* = 33)	4 (16.7%) Insufficient
ST11	Vande Castelee et al., 2015 [46]	Belgium	Randomized controlled clinical trial	Infliximab	251 IBD adult patients with a stable clinical response for at least 14 weeks (138/113, 41 (34.5–49.0) yo)	2 years and 16 weeks	Costs Cost savings QALY ICER	Proactive TDM (*n* = 128) vs. Empirical strategy (123)	18 (75.0%) Very good
ST12	Steenholdt et al., 2015 [47]	Denmark	Randomized controlled clinical trial	Infliximab	69 CD adult patients with LOR to infliximab (27/69, 37 (19–81) yo)	20 and 52 weeks	Costs Cost savings	Reactive TDM (*n* = 33) vs. Empirical strategy (*n* = 36)	17 (70.8%) Very good
ST13	Steenholdt et al., 2014 [48]					12 weeks			17 (70.8%) Very good

M: male; f: female; CHEERS: Consolidated Health Economic Evaluation Reporting Standards; CD: Crohn’s disease; QALY: quality-adjusted life-years; ICER: incremental cost-effectiveness ratio; IBD: inflammatory bowel disease; yo: years old; NA: not available; LOR: loss of response.

**Table 2 pharmaceutics-14-01009-t002:** Summary of the economic outcomes of each study per year.

Study	Authors, Year	Total Cost	Cost Savings	Average Cost Savings per Patient	QALY	ICER
ST1	Yao et al., 2021 [36]	PA: USD 110,851.18 (EUR 94,223.50) RA: USD 111,508.01 (EUR 94,781.81)	PA: USD 656.83 (EUR 558.31) compared to RA	PA: EUR 0.06 compared to RA	PA: 0.81 RA: 0.74	RA-PA: Dominated by PA
ST2	Negoescu et al., 2020 [37]	PA: USD 16,585.42 (EUR 14,926.88) RA: USD 15,847.69 (EUR 14,262.92) ES: USD 15,853.68 (EUR 14,268.31)	RA: USD 737.73 (EUR 663.96) compared to PA and USD 5.99 (EUR 5.39) compared to ES	NA	PA: 0.74 RA: 0.73 ES: 0.73	RA-PA: USD 146,509.12 (EUR 131,858.21) ES-RA: Dominated by RA
ST3	Attar et al., 2019 [38]	PA: EUR 186,635,650 ES: EUR 201,879,000	PA: EUR 15,243,350 compared to ES	PA: EUR 0.76 compared to ES	NA	NA
ST4	Freeman et al., 2016 [39]	PA: GBP 13,980 (EUR 18,174) ES: GBP 15,050 (EUR 19,565)	PA: GBP 1070 (EUR 1391) compared to ES	NA	PA: 0.63 ES: 0.65	ES-PA: GBP 43,727.01 (EUR 56,845.12)
ST5	Roblin et al., 2015 [40]	NA	RA: EUR 26,260,058.60 compared to ES	NA	NA	NA
ST6	Velayos et al., 2013 [41]	RA: USD 31,870 (EUR 23,902.5) ES: USD 37,266 (EUR 27,949.5)	RA: USD 5396 (EUR 4047) compared to ES	NA	RA: 0.80 ES: 0.80	ES-RA: Dominated by RA
ST7	Wu et al., 2021 [42]	NA	PA: AUD 304,916.95 (EUR 196,394.48) compared to RA	NA	NA	NA
ST8	Ganesnanthan et al., 2020 [43]	NA	PA: GBP 56,865 (EUR 62,551) compared to ES PA post-induction: GBP 51,595 (EUR 56,754.50) compared to ES RA: GBP 27,081.85 (EUR 29,790.04) compared to ES	NA	NA	NA
ST9	Guidi et al., 2018 [44]	RA: EUR 3,230,810.44 ES: EUR 3,788,285.67	RA: EUR 557,475.23 compared to ES	RA: EUR 39,197.38 compared to ES	NA	NA
ST10	Taks et al., 2017 [45]	NA	PA: EUR 47,026 compared to RA	NA	NA	NA
ST11	Vande Castelee et al., 2015 [46]	PA: EUR 5,201,473 ES: EUR 5,276,773	PA: EUR 75,300 compared to ES	PA: EUR 300 compared to ES	PA: 0.82 ES: 0.84	ES-PA: EUR 3,901,554.40
ST12	Steenholdt et al., 2015 [47]	RA: USD 22,066 (EUR 17,652.80) ES: USD 29,072 (EUR 23,257.60)	RA: USD 7006 (EUR 5604.8) compared to ES	RA: EUR 111.11 compared to ES	NA	NA
ST13	Steenholdt et al., 2014 [48]	RA: USD 26,164.67 (EUR 19,623.5) ES: USD 39,771.33 (EUR 29,828.5)	RA: USD 13,606.67 (EUR 10,205) compared to ES	RA: EUR 233.92 compared to ES	NA	NA

QALY: quality-adjusted life-years; ICER: incremental cost-effectiveness ratio; PA: proactive TDM; RA: reactive TDM; ES: empirical strategy; NA: not available; USD: United States Dollars; AUD: Australian dollars.

## Data Availability

Data are available from the corresponding author on reasonable request.

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
