# Peer review of "Cost-Effectiveness of Therapeutic Drug Monitoring of Anti-TNF Therapy in Inflammatory Bowel Disease: A Systematic Review"

_pharmaceutics, 2022, doi:10.3390/pharmaceutics14051009_

Round 1

Reviewer 1 Report

Overall, very well done.  Important topic.  

 It includes all the currently available information. It is well written and comprehensive.    Minor points:
  1. The discussion is long and rather descriptive. The parts describing the studies should be included in the 'results' section. 
  2. The data in the 'results' section should be moved to results.  Would group as proactive TDM vs. reactive TDM; proactive TDM vs. empirical strategy and reactive TDM vs. empirical strategy. 
  3. The data in the tables would be better described and grouped by 'Modeling approach' vs 'clinical studies'

Reviewer 2 Report

The authors did a great effort and invested a lot of time into this review and conducted an analytical cross-sectional analysis to see if therapeutic drug monitoring of anti-TNF therapy in inflammatory bowel disease was cost-effective. The following suggestions and comments may be valuable to the authors in improving the readability of their review.:

Detailed Comments and suggestions:

  • Please add a paragraph in the introduction regarding different Anti-TNF agents (They're also called TNF blockers, biologic therapies, or anti-TNF drugs). Clarify reasons for using Infliximab and Adalimumab only for your study and why you neglect other Anti-TNF agents such as Certolizumab, Etanercept, and Golimumab.

  • Regarding the previous point, if the authors broaden their research to include all these biological therapies, the number of particles may be increased to more than 108, please discuss that in the study design.

  • At the start of paragraphs, Line 69, 148….(This was cross…..This section may be…..), please edit these sentences to reflect your thoughts in a good manner and make sense for the readers.

  • The same at line 151, The systematic review identified a total of 102 publications…..Modify and paraphrase.

  • In 2.4. Final selection of articles,

The authors mentioned that (the selection of relevant articles was performed by (S.M.-M. and A.R.-L.) and a third reviewer (R.N.-M.) was responsible for reaching a resolution and subsequent consensus amongst all the authors), although in the author contributions section (Line 402)

They mentioned only S.M.-M. and A.R.-L. in the Methodology. The methodology was repeated two times in that section??? The second one includes (R.N.-M.). Please revise carefully and match between them.

  • Population in the included studies were variant and consequently, the cost variation is varied according to the no of patients included, my suggestion is to make another measure outcome of (cost saving per patient) and compare that to evaluate the pest Therapy plan and TDM procedures from the 13 studies included in that review. This will give insights to the readers of the best practices they can follow and implement in their associations (especially studies with AAA).

  • Table 1, please add a new column to the left containing the numbering of the studies included from 1-13. Then, add that numbers to the results and discussion part in order to make it easy for readers to follow and refer with keeping their citation numbers as it is.

  • Table 1, add a column mentioning the study concerned with Infliximab or Adalimumab or both of them.

  • Study 36 is the only one for adalimumab and the only study based on the pediatric population with the highest cost-saving and QALY, please discuss that and clarify the reasons to include it in the cross-sectional study.

  • The conclusion part does not reflect the main aims of the review article. The authors need to edit it and include the main results regarding cost reduction and benefits of TDM in Anti-TNF agent use. In addition, their recommendations for those who are dealing with these cases throughout the world.

  • In the supplementary file, please add heading titles for each column in Tables S2 and S3

Round 2

Reviewer 2 Report

Many thanks to the authors for their efforts and response to comments and suggetions.